:ᴐ᷄PLOS | ONE

# Evaluation of a savings-led family-based economic empowerment intervention for AIDS-affected adolescents in Uganda: A four-year follow-up on efficacy and cost-effectiveness

**Yesim Tozan**[1], **Sicong Sun**[2], **Ariadna Capasso**[1], **Julia Shu-Huah Wang**[3], **Torsten B. Neilands**[4], **Ozge Sensoy Bahar**[2], **Christopher Damulira**[2], **Fred M. Ssewamala**[2]*

1 College of Global Public Health, New York University, New York, New York, United States of America,
2 International Center for Child Health and Development, Brown School, Washington University in Saint Louis, Saint Louis, Missouri, United States of America, 3 Department of Social Work and Social Administration, The University of Hong Kong, Hong Kong SAR, China, 4 Center for AIDS Prevention Studies, School of Medicine, University of California, San Francisco, San Francisco, California, United States of America

* fms1@wustl.edu

## Abstract

### Background

Children who have lost a parent to HIV/AIDS, known as AIDS orphans, face multiple stressors affecting their health and development. Family economic empowerment (FEE) interventions have the potential to improve these outcomes and mitigate the risks they face. We present efficacy and cost-effectiveness analyses of the Bridges study, a savings-led FEE intervention among AIDS-orphaned adolescents in Uganda at four-year follow-up.

### Methods

Intent-to-treat analyses using multilevel models compared the effects of two savings-led treatment arms: Bridges (1:1 matched incentive) and BridgesPLUS (2:1 matched incentive) to a usual care control group on the following outcomes: self-rated health, sexual health, and mental health functioning. Total per-participant costs for each arm were calculated using the treatment-on-the-treated sample. Intervention effects and per-participant costs were used to calculate incremental cost-effectiveness ratios (ICERs).

### Findings

Among 1,383 participants, 55% were female, 20% were double orphans. Mean age was 12 years at baseline. At 48-months, BridgesPLUS significantly improved self-rated health, (0.25, 95% CI 0.06, 0.43), HIV knowledge (0.21, 95% CI 0.01, 0.41), self-concept (0.26, 95% CI 0.09, 0.44), and self-efficacy (0.26, 95% CI 0.09, 0.43) and lowered hopelessness (-0.28, 95% CI -0.43, -0.12); whereas Bridges improved self-rated health (0.26, 95% CI

**Data Availability Statement:** All relevant data are within the manuscript and its Supporting Information files.

**Funding:** Funded by FMS. Grant #1 R01-HD070727-01. National Institute of Child Health and Development at the National Institutes of Health, https://www.nichd.nih.gov/. The funders had no role in study design, data collection and analysis, decision to publish, or preparation of the manuscript.

**Competing interests:** The authors have declared that no competing interests exist.

0.08, 0.43) and HIV knowledge (0.22, 95% CI 0.05, 0.39). ICERs ranged from $224 for hopelessness to $298 for HIV knowledge per 0.2 standard deviation change.

## Conclusions

Most intervention effects were sustained in both treatment arms at two years post-intervention. Higher matching incentives yielded a significant and lasting effect on a greater number of outcomes among adolescents compared to lower matching incentives at a similar incremental cost per unit effect. These findings contribute to the evidence supporting the incorporation of FEE interventions within national social protection frameworks.

## Introduction

Healthy adolescence is a critical step in transitioning to a healthy adulthood [1]. This developmental period can, however, be thwarted by poverty and lack of access to healthcare, education and social support. Most health risks and challenges, including exploratory sexual behaviors, substance use, and poor mental health functioning, emerge during adolescence [2]. Orphaned adolescents, defined as those who have lost one or both biological parents, face additional economic and psychosocial barriers that further limit their ability to thrive without appropriate support systems [3, 4].

Globally, over 12 million children under age 18 are reported to have lost at least one parent to HIV/AIDS [5]. Of these, 79% live in sub-Saharan Africa (SSA) [5]. Within SSA, Uganda is heavily affected; of the 1.9 orphaned children, those orphaned by AIDS make up 35% (660,000) [6]. Research shows that AIDS orphaned children are more likely than other orphans to initiate sex early [7], report higher rates of transactional sex [8], engage in risky sexual behaviors [9], test positive for HIV infection [10, 11], have higher vulnerability to violence and abuse [3, 8], and report poorer mental health [10, 11]. Poverty mediates all of these effects [8]. Parental loss has also been linked to increased household responsibilities and reduced school attendance and performance in this age group [12].

While social and emotional support promote coping and resilience among adolescent orphans, which is key to leading a healthy and productive life [3, 13], household financial instability is associated with poor educational and health outcomes in this population [14–16]. Theory and evidence suggest that social protection programs can mitigate risks among children affected by HIV/AIDS [14, 17]. In particular, cash assistance may reduce sexual risk taking, including transactional sex, thus reducing new HIV infections, unwanted pregnancies and school dropout rates among adolescents [6, 18]. Grounded in asset theory, family economic empowerment (FEE) interventions go beyond cash transfers by providing capacity building, mentorship, and seed funding and fostering household financial stability through promoting income generating activities and financial literacy [19, 20]. Asset theory posits that orphaned adolescents will experience higher levels of depression, have worse educational outcomes, and engage in higher risk behaviors if they lack financial means to participate in secondary education. FEE interventions that bring financial stability to households act as a protective factor [21]. In fact, these interventions have been shown to increase household financial stability, leading to greater savings for education and improved academic performance and mental health outcomes (lower depressive symptomatology and higher levels of self-esteem) among orphaned adolescences in Uganda [16, 20].

FEE interventions offer promise in addressing several health and developmental needs for poor adolescents impacted by HIV/AIDS in low-resource communities, including SSA [20, 22–25]. To make the case for public investment in these interventions, it is necessary to integrate the evidence on effects with costs. By calculating an incremental cost per unit of benefit, cost-effectiveness analyses make it explicit which interventions will contribute the most relative to their costs and inform resources allocation decisions in the face of competing health priorities and resource constraints. Evidence is extremely limited on the cost-effectiveness of interventions aimed at improving adolescents' health outcomes, including physical and mental health. This paper contributes to the currently limited scientific body of knowledge on the economic value of an FEE intervention, titled Bridges, that applies a savings-led approach aimed at improving health and developmental outcomes of poor adolescents. Specifically, our earlier study findings showed that the Bridges intervention demonstrated a favorable effect on the critical developmental outcomes of adolescents impacted by HIV/AIDS compared with usual care alone at 24-months post-intervention initiation [26]. A longer-term sustainability question has, however, remained unanswered by those initial short-term findings: how long would the observed outcomes be sustained and at what cost? That is the question addressed in this paper, using data from 48 months post-intervention initiation. If the intervention exhibits sustained efficacy, it is also important to understand the potential changes in the cost-effectiveness of the intervention over time. In summary, this analysis aims to assess the effects and cost-effectiveness of the two arms of a FEE intervention in relation to care-as-usual two years post-intervention.

## Materials and methods

### Trial population and setting

The current analysis is based on baseline and 48-month follow-up data from the Bridges to the Future intervention (hereinafter, Bridges). Bridges was a five-year (2012–2016) cluster randomized control trial to evaluate the efficacy and cost-effectiveness of a FEE intervention on health, developmental and educational outcomes among adolescents orphaned by HIV/AIDS in southwestern Uganda. All procedures performed in studies involving human participants were in accordance with the ethical standards of the institutional and national research committee and with 1964 Helsinki declaration and its later amendments. Informed consent and assent were obtained separately from caregivers and adolescents respectively, prior to study participation. The Bridges study received approval from the Columbia University Institutional Review Board (IRB-AAA11950) and the Uganda National Council of Science and Technology (SS2586). The study is registered in the Clinical Trials database (NCT01447615). The clinical trial is described in detail elsewhere [26]. Briefly, a total of 1,410 adolescents orphaned by HIV/AIDS (n = 621 boys, n = 789 girls), between the ages of 10–16 (mean participant age: 12 years at baseline) were recruited from 48 public primary schools (Table A in S1 Appendix). The schools are in four political districts of Rakai, Masaka, Lwengo and Kalungu in southwestern Uganda—a region heavily affected by HIV/AIDS [27]. There were three eligibility criteria: 1) the adolescent had lost one or both parents to HIV/AIDS; 2) the adolescent was enrolled in grades 5 or 6, in a government-aided primary school during recruitment period; and 3) the adolescent was living with a family, not in an institution. A total of 27 adolescents were deemed ineligible after randomization and dropped from the trial, resulting in a final sample of 1,383 adolescents (Fig. A in S1 Appendix). About 20% of the adolescents were double orphans (lost both parents). Schools were randomly assigned to three study arms: control, Bridges, and BridgePLUS (described below).

**Table 1. Itemized total per-participant costs (all costs are in 2012 Ugandan Shillings unless otherwise indicated).**

| | Usual Care | Bridges | BridgesPLUS |
|---|---|---|---|
| **Recurrent costs** | | | |
| School lunches | 39,491 | 39,491 | 39,491 |
| Educational materials | 76,569 | 76,569 | 76,569 |
| Counseling | 38 | 38 | 38 |
| Recruitment of participants | 5,847 | 5,847 | 5,847 |
| Child savings account | | | |
| *Account opening* | - | 27,414 | 27,414 |
| *Initial account deposit* | - | 20,000 | 20,000 |
| *Annual matched savings* | - | 20,309 | 42,634 |
| Mentorship | - | 55,490 | 55,490 |
| Financial education and income generating activity training | - | 53,710 | 53,710 |
| Personnel (staff salary and allowances) | 100,105 | 200,210 | 200,210 |
| Volunteer and donated resources | 5,986 | 429,526 | 429,525 |
| **Capital costs** | | | |
| Furniture, equipment and vehicles | 66,285 | 132,570 | 132,570 |
| **Total per-participant costs** | **294,321** | **1,061,174** | **1,083,498** |
| **Total per-participant costs (in 2012 USD)** | **117** | **419** | **428** |

Adolescents in the control condition received 'usual care' consisting of counseling by community priests and school supplies. Adolescents in Bridges and BridgesPLUS treatment arms received usual care plus an incentivized savings account [Child Development Account (CDA)] with either a 1:1 match rate (Bridges) or 2:1 match rate (BridgesPLUS). In addition, all participants in Bridges and BrigesPLUS received: three sessions on financial literacy and management (FLT), including how to save, budget and support asset accumulation; six sessions on income generating activities; and eight sessions of mentorship by a near peer. The first two activities were conducted by trained research assistants. The only difference between Bridges and BridgesPLUS was the level of incentivized match rate. The participants did not need to attend all the activities to receive incentives. To be eligible, they needed to attend the FLT sessions and open a bank account at a participating financial institution. Matched funds were restricted for use on educational or income-generating activities. The matched funds were contributed by the trial (See Table 1 for program costs, including matched amounts). The attrition rate was 8.8% for the Bridges and the control groups and 11.2% for the BridgesPLUS group at the 48-month follow-up. However, the results from a chi-square test indicate that the attrition rates do not differ significantly by study arms ($\chi^2 = 2.04$; $p = 0.36$) (Table B in S1 Appendix).

## Measures

Self-rated health is measured by a single 5-item scale ranging from excellent to very poor with higher values indicate better health [28]. Mental health functioning is conceptualized as depression, hopelessness, self-concept and self-efficacy. We used the 27-item Child Depression Inventory (CDI) and the 20-item Beck Hopelessness Scale (BHS) [29] operationalized with higher scores indicating worse mental health. Moreover, the 20-item Tennessee Self-Concept Scale [30], and the 29-item Youth Self-Efficacy Survey [31] are used for measurements of self-concept and self-efficacy with higher values on indicate more positive self-concept and self-efficacy, respectively. Three indictors—sexual risk-taking intentions [20], HIV prevention intention [32], and HIV knowledge [21] are used to capture sexual health. Detailed example

questions for each scale are provided in Table C in S1 Appendix with reliability statistics. Briefly, CDI ($\alpha = 0.68$) and BHS ($\alpha = 0.65$) show modest internal consistency, whereas Tennessee Self-Concept Scale, the Youth Self-Efficacy Survey, sexual risk-taking intentions, HIV prevention intention, and HIV knowledge all have acceptable internal consistency with Cronbach's alphas greater than 0.70. All measurements mentioned above are standardized to be comparable across all outcomes.

## Evaluation of the intervention

Outcome data were collected by trained research assistants through in-person interviews with adolescents. To examine the intervention effects, multilevel linear regressions were conducted independently for each outcome using the final sample of 1,383 adolescents. We focused on eight key health and mental health outcomes: self-rated health, depression, hopelessness, self-concept, self-efficacy, sexual risk-taking intentions, HIV prevention attitudes, and HIV knowledge (Table C in S1 Appendix). We used a three-level multilevel model that accounted for school and individual clustering-effects by including school ID and child ID as random intercept terms. Further, we allowed the slopes over time to differ for each child. Scores were standardized before being included in the regression models to facilitate comparison of the effects across outcomes with different units. There were no significant differences in distributions across arms in students' age and sex at baseline. Therefore, these variables were not included in the models. In each model, we included study group dummies (Bridges and BridgesPLUS; usual care as the reference group), a wave dummy (48-month follow-up; baseline as the reference group), and their interactions. The interaction between 48-month follow-up indicator and treatment arms were the key coefficients of interest by demonstrating the effect of the intervention with baseline differences and trends in outcomes controlled for. After each multilevel regression, we used a Wald test to examine whether the coefficients between the two interaction terms were statistically different. All analyses were performed using the mixed command in Stata 15 (StataCorp LP, College Station, TX, USA).

## Costs of the intervention

We estimated the costs of the usual care and the two treatment arms from a provider perspective and used a combination of activity-based and ingredients approach to costing where we identified all the activities in each study arm and then measured and valued all the resources used for each activity in each study arm. Activities encompassed recruitment of participants, provision of school lunches and educational materials (uniforms, notebooks and textbooks), counseling, mentorship, financial education and income generating activity trainings, and opening and contributing to child savings accounts. In addition to actual financial expenditures, we also quantified volunteered and donated resources used for all the activities to arrive at the economic costs of the intervention. Costs were carefully recorded throughout the Bridges implementation process and extracted for analysis from the project's administrative records and CDA-related bank records. The recurrent costs of implementation included the costs of school supplies (students' lunches, school uniforms and textbooks), bank accounts opening, initial accounts deposits, matching incentive contributions, transportation (fuel, travel allowances and rental fees), maintenance and repair of equipment and vehicles, field office rent, office maintenance (Internet access and phone, utilities, security services), office supplies and printing, field personnel (staff salary and allowances), time costs of volunteers (community leaders and mentors) and other implementing partners (teachers and bank staff), and donated resources (classroom space). Time costs of staff and teachers were apportioned according to time devoted to the activities in each arm. The capital costs included the costs of office

furniture and equipment and vehicles. We calculated annual depreciation costs of the capital items assuming an appropriate useful life for each item and apportioned these costs according to their estimated share of use for the activities in each arm. We calculated the per-participant costs for each cost category using the treatment-on-the-treated (TOT) sample and added the costs to estimate the total per-participant cost for each arm over the intervention period. All costs were adjusted for inflation using the Uganda Consumer Price Index [33], discounted at 3% to the first year of the trial [34], and expressed in 2012 US dollars.

## Cost-effectiveness methodology

This cost-effectiveness analysis used the costing data collected during the Bridges study planning phase, baseline, through study implementation to closeout. The outcome data comes from data collected at baseline and at 48-month follow-up. The analysis centered on incremental cost-effectiveness ratios (ICERs), where the numerator represented the cost difference between the treatment arms and the usual care, and the denominator represents the difference in average treatment effects. To that end, the cost-effectiveness analysis of the Bridges intervention involved examining how much Bridges or BridgesPLUS costs to achieve a unit of effect relative to usual care. First, we calculated the total per-participant costs in each study arm. Because the intent-to-treat (ITT) sample is larger than those who actually received the intervention, we calculated the per-participant costs conservatively based on the TOT sample. The use of ITT sample does not affect the relative difference in incremental costs across the two intervention arms and hence the relative difference in the cost-effectiveness ratios. Second, we estimated the effects of the intervention on our outcomes of interest as the standardized mean difference between each of the two treatment arms and the usual care arm, known as effect sizes, using an ITT approach. For the outcomes, we chose 0.2 SD change as a threshold, which corresponds to a small effect size as per guidelines provided by Cohen [35]. Third, we calculated the ICERs using the formulation above and computed the per-participant cost per 0.2 SD change for each outcome. Reporting of this analysis followed the Consolidated Health Economic Evaluation Reporting Standards (CHEERS) [36].

## Sensitivity analysis

In sensitivity analysis, we used the 95% confidence interval (CI) of the effect sizes and varied the total per-participant costs of the intervention from 80% to 120% and re-calculated the ICERs over these cautious ranges so as to explore the pessimistic (high cost/low effectiveness) and optimistic (low cost/high effectiveness) scenarios for Bridges and BridgesPLUS [37].

## Results

### The effects of the intervention

Table 2 present the intervention effects on the key health and mental health outcomes from multilevel linear regression analyses. The intervention increased self-rated health among participants in both treatment arms as compared to those receiving usual care (Bridges: 0.26 SD, 95% CI 0.08, 0.43; BridgesPLUS: 0.25 SD, 95% CI 0.06, 0.43). Participants in both Bridges and BridgesPLUS gained in HIV knowledge over participants in the control arm (0.22 SD, 95% CI 0.05, 0.38 and 0.21 SD, 95% CI 0.01, 0.41, respectively). Only BridgesPLUS resulted in statistically significant lower levels of hopelessness (-0.28 SD, 95% CI -0.43, -0.12), and higher levels of self-concept (0.26 SD, 95% CI 0.09, 0.44), and self-efficacy (0.26 SD, 95% CI 0.09, 0.43) as compared to the usual care arm. There are no observable statistically significant intervention effects on levels of depression, sexual risk-taking intentions, and HIV prevention attitudes at

**Table 2. Intervention effects on self-rated health, mental health functioning, and sexual health.**

| Outcome | Self-rated health (n = 1,383) | | Depression (n = 1,382) | | Hopelessness (n = 1,382) | | Self-concept (n = 1,383) | | Self-efficacy (n = 1,383) | | Sexual risk-taking intention (n = 1,383) | | HIV prevention attitudes (n = 1,383) | | HIV knowledge (n = 1,383) | |
|---|---|---|---|---|---|---|---|---|---|---|---|---|---|---|---|---|
| | Effect size (95% CI) | p-value | Effect size (95% CI) | p-value | Effect size (95% CI) | p-value | Effect size (95% CI) | p-value | Effect size (95% CI) | p-value | Effect size (95% CI) | p-value | Effect size (95% CI) | p-value | Effect size (95% CI) | p-value |
| **Group (ref: usual care)** | | | | | | | | | | | | | | | | |
| Bridges | -0.19 (-0.34 to -0.04) | 0.011 | -0.01 (-0.17 to 0.16) | 0.942 | -0.08 (-0.21 to 0.04) | 0.177 | 0.002 (-0.14 to 0.14) | 0.982 | -0.08 (-0.24 to 0.07) | 0.293 | 0.02 (-0.14 to 0.17) | 0.849 | 0.09 (-0.11 to 0.29) | 0.381 | 0.01 (-0.20 to 0.22) | 0.948 |
| Bridges PLUS | -0.16 (-.32 to -0.01) | 0.041 | 0.07 (-0.12 to 0.25) | 0.496 | 0.04 (-0.11 to 0.19) | 0.628 | -0.20 (-0.34 to -0.06) | 0.006 | -0.20 (-0.41 to 0.0001) | 0.050 | 0.01 (-0.14 to 0.15) | 0.948 | 0.00 (-0.17 to 0.18) | 0.979 | -0.03 (-0.26 to 0.21) | 0.828 |
| **Time (ref: baseline)** | | | | | | | | | | | | | | | | |
| 48 months | -0.15 (-0.28 to -0.02) | 0.027 | -0.22 (-0.33 to -0.10) | <0.001 | -0.51 (-0.59 to -0.43) | <0.001 | 0.39 (0.25 to 0.52) | <0.001 | 0.37 (0.25 to 0.49) | <0.001 | -0.07 (-0.20 to 0.06) | 0.297 | 0.81 (0.64 to 0.99) | <0.001 | 0.77 (0.63 to 0.91) | <0.001 |
| **Group X time** | | | | | | | | | | | | | | | | |
| Bridges X 48 months | 0.26 (0.08 to 0.43) | 0.01 | -0.10 (-0.31 to 0.11) | 0.337 | -0.10 (-0.25 to 0.06) | 0.220 | 0.03 (-0.16 to 0.22) | 0.752 | 0.02 (-0.16 to 0.20) | 0.818 | -0.17 (-0.38 to 0.03) | 0.097 | -0.12 (-0.34 to 0.10) | 0.273 | 0.22 (0.05 to 0.38) | 0.012 |
| Bridges PLUS X 48 months | 0.25 (0.06 to 0.43) | 0.03 | -0.14 (-0.32 to 0.03) | 0.098 | -0.28 (-0.43 to 0.13) | <0.001 | 0.26 (0.09 to 0.44) | 0.003 | 0.26 (0.09 to 0.43) | 0.003 | -0.06 (-0.22 to 0.10) | 0.465 | -0.04 (-0.25 to 0.16) | 0.679 | 0.21 (0.01 to 0.41) | 0.04 |
| **Wald test: Bridges X 48 months = Bridges PLUS X 48 months ($\chi^2$)** | 0.01 | 0.92 | 0.16 | 0.69 | 3.71 | 0.054 | 7.1 | 0.008 | 7.06 | 0.008 | 1.5 | 0.221 | 0.92 | 0.336 | 0.01 | 0.934 |

Note: One child in the control group, and one child in the Bridges PLUS were found to have missing values for both baseline and 48-months, in term of depression and hopelessness, respectively, thus they were excluded in the analysis for the corresponding outcome.

four-year follow-up. Wald tests point to statistically significant differences in efficacy of the treatment arms over time on self-concept ($\chi^2$ = 7.10, $p$ = 0.008) and self-efficacy ($\chi^2$ = 7.06, $p$ = 0.008), and a moderate though not statistically significant difference on hopelessness ($\chi^2$ = 3.71, $p$ = 0.054).

## The costs and cost-effectiveness of the intervention

Table 1 presents the total per-participant costs for the three study arms. The total per-participant cost was $117 for the usual care arm, $419 for the Bridges arm, and $428 for the BridgesPLUS arm. The observed very small difference in cost between Bridges and BridgesPLUS was a result of the match rate difference.

Table 3 presents the ICERs for the outcomes on which the effects of Bridges and BridgesPLUS were found to be statistically significant relative to usual care. The ICER for self-rated health for Bridges was US$236 (95% CI 139–766) per 0.2 SD and lower than for BridgesPLUS at US$252 (95% CI 144–1014) per 0.2 SD, reflecting the higher mean effect size given the very small difference in the total per-participant costs of the interventions relative to usual care. Precisely for the same reasons, the ICER for HIV knowledge was lower at US$279 (95% CI 157–1,235) per 0.2 SD for Bridges compared to US$298 (95% CI 153–6,933) per 0.2 SD for BridgesPLUS. The intervention effects on hopelessness, self-concept and self-efficacy were statistically significant only for the BridgesPLUS arm, and the ICERs were computed as US$224 (95% CI 145–499), US$236 (95% CI 143–686), and US$242 (95% CI 145–734) per 0.2 SD,

**Table 3. Incremental cost-effectiveness ratios (ICER) relative to usual care (in 2012 US dollars).**

| Outcome | Bridges ICER (95% CI) | BridgesPLUS ICER (95% CI) |
|---|---|---|
| Self-rated health | 236 (139–766) | 252 (144–1,040) |
| Hopelessness | *624 (n.s.)* | 224 (145–499) |
| Self-concept | *1,952 (n.s.)* | 236 (143–686) |
| Self-efficacy | *288 (n.s)* | 241 (145–734) |
| HIV knowledge | 279 (157–1,235) | 298 (153–6,933) |

Note: Unit represents a 0.2 standard deviation change in mean outcome. Italics indicate non-significance.

respectively. Overall, BridgesPLUS had a statistically significant effect on a higher number of health and mental health outcomes compared to Bridges, with ICERs ranging between US $224–298 per 0.2 SD change.

Table 4 shows the results of the sensitivity analysis and presents the ICERs for the same set of mental health and health related outcomes under the optimistic and pessimistic scenarios, corroborating our findings that Bridges cost less than BridgesPLUS to achieve a 0.2 SD change in the outcomes for which the effects were statistically significant for both treatment arms at four-year follow-up.

## Discussion

In this study, we examined the efficacy and cost-effectiveness of a savings-led FEE intervention targeted at adolescents orphaned by HIV/AIDS in Uganda. We focused on eight different health and mental health outcomes, namely self-rated health, depression, hopelessness, self-concept, self-efficacy, sexual risk-taking intentions, HIV prevention attitudes, and HIV knowledge. At 24 months post intervention initiation [26], adolescents receiving Bridges (lower saving incentive) and BridgesPLUS (higher saving incentive) reported similar outcomes in regards to health and mental health, self-concept, self-efficacy, and HIV knowledge. Specifically, when compared to those in usual care, adolescents in both treatment conditions did better, although a higher savings incentive did not lead to differential treatment effects and cost-effectiveness ratios between the two treatment conditions [26]. At 48-month follow-up, the effects of Bridges and BridgesPLUS on self-rated health and HIV knowledge remained statistically significant relative to usual care, while Bridges proved to be on average more cost-effective compared to BridgesPLUS with lower ICERs for these outcomes to attain a 0.2 SD change relative to usual care. From 24-month to 48-months (the post-intervention period), the effects

**Table 4. Sensitivity analysis of incremental cost-effectiveness ratios (ICERs) (in 2012 US dollars).**

| Outcome | Bridges | | BridgesPLUS | |
|---|---|---|---|---|
| | Optimistic scenario (Low cost/High effectiveness) | Pessimistic scenario (High cost/Low effectiveness) | Optimistic scenario (Low cost/High effectiveness) | Pessimistic scenario (High cost/Low effectiveness) |
| Self-rated health | 112 | 919 | 115 | 1,246 |
| Hopelessness | - | - | 116 | 598 |
| Self-concept | - | - | 114 | 822 |
| Self-efficacy | - | - | 116 | 880 |
| HIV knowledge | 126 | 1,482 | 122 | 8,307 |

Note: Unit represents a 0.2 standard deviation change in mean outcome.

on hopelessness, self-concept and self-efficacy became statistically insignificant for adolescents receiving Bridges (lower saving incentive), while BridgesPLUS (higher saving incentive) significantly improved these outcomes, with ICERs ranging between US$224–298 per 0.2 SD change. These ICERs are overall higher than those calculated at 24-month follow-up (US$166–263 per 0.2 SD change) [26] because of the reduced effects of the intervention at 48-month follow-up. Both treatment conditions had no observable effects on HIV prevention attitudes, depression and sexual risk-taking intentions at 48-month follow-up. Thus, these specific outcomes were not included in the cost-effectiveness analysis. However, it is noteworthy to point out that, even though not statistically significant, the changes in depression and sexual risk-taking intentions were in the expected direction in both treatment conditions.

We calculated the per-participant costs conservatively based on the TOT sample. As a comparison, the total per-participant costs based on the ITT sample would be lower at $103, $363, and $372, for the usual care, Bridges and BridgesPLUS arms, respectively (Table D in S1 Appendix) and would result in lower ICERs per 0.2 SD change relative to usual care. Given the resource-intensive implementation strategy used in delivering the Bridges FEE intervention, and the fact that the cost analysis presented here used a conservative costing approach using the TOT sample, the cost per participant is likely to decrease when the intervention is integrated into a broader healthcare system. It is also likely that there may be economies of scale that the intervention would benefit from when it is delivered at scale.

The Bridges study targeted multiple outcomes for adolescents, and we presented the ICERs individually for each outcome, highlighting the important differences across the two treatment arms over time. Our findings indicate that Bridges and BridgesPLUS continued to have a positive effect on several health and mental outcomes, including self-rated health and HIV knowledge relative to usual care at 48-month follow-up, with ICERs ranging between US$236–298 per 0.2 SD change. An especially noteworthy finding of this follow-up is that the higher matching rate in BridgesPLUS yielded a significant and lasting effect on multiple outcomes in the long term, namely adolescents' self-rated health, hopelessness, self-concept, and self-efficacy compared to Bridges at a similar incremental cost per unit effect, whereas the effects on adolescents who received the lower matching rate faded over the same period.

Cash transfer programs, both conditional and unconditional, targeted at most vulnerable households have been shown to impact a multiplicity of education and health outcomes in low resource-settings [38]. In Malawi, the government-run cash transfer program targeting ultra-poor, labor-constrained households increased school enrollment and reduced drop-out rates and improved mental health outcomes in children after one year [39]. In Kenya, the government-led program for orphans and vulnerable children, aimed at reducing HIV risk through unconditional cash transfers, reduced the odds of sexual debut by 31% [40]. Structural interventions to reduce HIV incidence have been found to be the most cost-effective [41], particularly when targeting multiple outcomes, such as education, in addition to HIV/AIDS incidence [42]. A privately funded program in Malawi conditioned on continued school enrollment of adolescent girls reduced the prevalence of HIV, Herpes simplex virus and depression and improved school attendance in this population. At 18 months, the intervention was estimated to avert 209 disability-adjusted life years (DALYs) at a cost of $297 per DALY averted [43]. Despite the growing literature on cash transfer programs, few studies have focused on interventions that include an economic empowerment component [41].

Our findings make a unique contribution to the existing literature on incentivized savings as previous studies on the effectiveness of savings incentives have primarily focused on economic outcomes [44–46]. Given the extremely limited cost-effectiveness evidence base on savings-led FEE interventions, there are no benchmarks associated with these intermediate health and mental outcomes, which limits our interpretation of the cost-effectiveness results.

Generally speaking, the issue of what an additional unit of outcome is worth needs to be addressed for these interventions. As we wait for the empirical evidence that describes the longer-run, final outcomes such as the incidence of HIV or other sexually transmitted diseases becomes available, this issue can be best addressed with accumulating evidence from cost-effectiveness analyses of these interventions in similar settings and populations by examining if the same effect can be achieved at a lower cost, presuming that the effects are measured in the same units [47, 48]. In this pioneering study, we based and compared and contrasted our findings on a rigorous analysis of the efficacy and cost data relating to the two treatments arms with differing incentivized match rates at 24- and 48-month post-intervention initiation.

Overall, our findings suggest that this multifaceted intervention has the potential to positively contribute, both in the short-term and long-term, to the health and overall development of adolescents impacted by HIV/AIDS by mitigating household financial instability. Further, the rate of incentive seems to matter to sustain the effects of the intervention in the long run. We believe that the significant effects sustained on 'multiple' critical outcomes in this high-risk population justify investments in savings-led economic empowerment interventions in resource limited-settings.

A limitation of this study is that multiple hypotheses were tested for multiple outcomes using the same exposure and covariate data, potentially leading to inflated Type 1 error rates. Future studies seeking to replicate our findings might consider concentrating on the subset outcomes found to be statistically significant in this study and employing multiplicity adjustments to further control Type 1 error rates. Another limitation is that because our study was quantitative by design, we did not have qualitative data on adolescent perceptions of the program, which could have added nuance and context to better elucidate the mechanisms of the effectiveness of the program on youth. Future qualitative research is needed to understand the causal pathways of effectiveness of such family economic strengthening programs. Use of self-reported measures is the most common approach to assessment in low resource settings, but this approach is limited by self-report bias [49]. This study also relied on self-reported measures for physical and mental health and sexual risk-taking behavior. Therefore, our findings are limited to self-reported changes in these outcomes rather than a clinical diagnosis, for instance, for mental health outcomes. Future studies should consider incorporating biomarkers and other objective measures. Further research is also warranted to replicate and extend these findings in other similar settings and help establish cost-effectiveness benchmarks that can be useful for researchers in intervention development and for policymakers in decision-making. Accumulating evidence will further strengthen confidence in the benefits and economic value of savings-led and incentivized economic empowerment interventions for vulnerable adolescents living in countries with limited resources.

## Supporting information

**S1 Appendix.** Fig. A. CONSORT Flow Diagram: *The Bridges to the Future Study* (2011–2017) Table A. Descriptive statistics of adolescent characteristics at baseline Table B. Descriptive statistics on characteristics of attrited sample Table C. Description of outcome measures Table D. Itemized total per-participant costs based on intent-to-treat sample (all costs are in 2012 Ugandan Shillings unless otherwise indicated).
(DOCX)

## Author Contributions

**Conceptualization:** Yesim Tozan, Ozge Sensoy Bahar, Fred M. Ssewamala.

**Data curation:** Julia Shu-Huah Wang.

**Formal analysis:** Sicong Sun, Ariadna Capasso.

**Investigation:** Christopher Damulira.

**Methodology:** Yesim Tozan, Ariadna Capasso, Julia Shu-Huah Wang, Torsten B. Neilands.

**Project administration:** Christopher Damulira.

**Resources:** Fred M. Ssewamala.

**Supervision:** Fred M. Ssewamala.

**Writing – original draft:** Ariadna Capasso.

**Writing – review & editing:** Yesim Tozan, Julia Shu-Huah Wang, Ozge Sensoy Bahar, Fred M. Ssewamala.

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
