## [Decision Letter · Decision Letter 0]

5 Sep 2019

PONE-D-19-20904

Evaluation of a savings-led family-based economic empowerment intervention for AIDS-affected adolescents in Uganda: A four-year follow-up on efficacy and cost-effectiveness

PLOS ONE

Dear Dr. Ssewamala,

Thank you for submitting your manuscript to PLOS ONE. After careful consideration, we feel that it has merit but does not fully meet PLOS ONE’s publication criteria as it currently stands. Therefore, we invite you to submit a revised version of the manuscript that addresses the points raised during the review process.

The reviewers provide useful comment and suggestion for improvement. Please ensure you address all comments related to the methods and interpretation of results.   

We would appreciate receiving your revised manuscript by Oct 20 2019 11:59PM. To enhance the reproducibility of your results, we recommend that if applicable you deposit your laboratory protocols in protocols.io, where a protocol can be assigned its own identifier (DOI) such that it can be cited independently in the future. For instructions see: http://journals.plos.org/plosone/s/submission-guidelines#loc-laboratory-protocols

We look forward to receiving your revised manuscript.

Kind regards,

Christopher M Doran, BEc (Hons) PhD

Academic Editor

PLOS ONE

Journal Requirements:

1. Please update your Ethics Statement on the online submission form to include the information on consent provided in the methods section of your manuscript.

2. There is some discrepancy between your manuscript and the details registered at https://clinicaltrials.gov/ct2/show/NCT01447615. Please clarify whether the ages eligible for study were 11-17 as stated at www.clinicaltrials.gov or 10-16 years as stated in your manuscript.

3. Data availability issue. In your statement you say "All relevant data are within the paper and its Supporting Information files", but as we explain in http://journals.plos.org/plosone/s/data-availability#loc-faqs-for-data-policy you should provide the individual data points behind means, medians and variance measures presented in the results, tables and figures, and not just those summary statistics. Please provide these underlying participant-level data in a supporting information file or public repository, taking care not to include identifying information (see http://www.bmj.com/content/340/bmj.c181.long); if these data cannot be publicly deposited or included in the supporting information, e.g. due to patient privacy, legal reasons, or being provided by a third party, please explain why and explain how researchers may access them. Note that authors should not be the sole named individuals responsible for ensuring data access.

Reviewers' comments:

Reviewer's Responses to Questions

**Comments to the Author**

1. Is the manuscript technically sound, and do the data support the conclusions?

Reviewer #1: Yes

Reviewer #2: Yes

Reviewer #3: Partly

2. Has the statistical analysis been performed appropriately and rigorously? 

Reviewer #1: Yes

Reviewer #2: Yes

Reviewer #3: I Don't Know

3. Have the authors made all data underlying the findings in their manuscript fully available?

Reviewer #1: Yes

Reviewer #2: No

Reviewer #3: Yes

4. Is the manuscript presented in an intelligible fashion and written in standard English?

Reviewer #1: Yes

Reviewer #2: Yes

Reviewer #3: No

5. Review Comments to the Author

Reviewer #1: The manuscript is generally well written and presented. However there some minor limitations for publishing.

1. The core question being addressed is not specifically stated or clearly linked to the extensive analyses provided. This could be due to the fact that the authors assume the audience is clear in regards to the intent of the trail, as well as of he actual project itself. The authors would greatly improve this issue by providing the readers with both a slightly elaborated description of the overall project, more on the underlying rationale for a focus on cost-effectiveness of such projects in global low income and rural settings, especially those with a history of social and cultural upheaval.

2. The retrospective focus of the analyses does not take into account the contextual and process variables that have also influenced the observed outcomes. This is one limitation of a single paradigm (quantitative) focus and self-reported behavioral outcomes. The authors need to better address the potential influence of these factors - what was provided was highly inadequate.

3. In such project the personnel and setting are important to any analyses since both are critical to the functional environments (e.g, relationship with youth, physical settings, so forth). While recognized as not a part of the present study, some mention of contextual variables was both expected and needed.

4. Lastly, operational definitions of the key variables of health and others were not provided. This needs to be addressed.

Overall, a good study. If the limitations are addressed this manuscript will provide a means to contribute our knowledge of such projects as suggested.

Reviewer #2: I have made comments and raised queries in the attached track-changed word documents.

However I would like to add two specific things here that I'd expect the authors to address 1) They need to note that as shown in the supplementary tables specifically Table S1 there was a marked higher attrition by 48 months for the BridgesPlus group (11.2%) as opposed to both Bridges and Control (8.8%) groups compared to the numbers at baseline. 2) There should be some mention of the fact that they are using the same exposure and co-variate data to test multiple hypotheses and some readers may wonder whether there is a multiplicity problem here requiring a lower P-value to be used. Both of these should be highlighted in a small limitations sub-section within the discussion.

Reviewer #3: The investigators have presented the long-term (48 months) health-related and cost outcomes for an economic intervention with HIV/AIDs related orphans. The findings have the potential to contribute significantly to the evidence on savings-led programs with adolescents and sustained impact and cost. I have several suggestions for improvement. The methods section is lacking in important details related to setting, sample, self-report measures (psychometrics, reliability of measures) used with adolescents, including data collection procedures, and brief description (perhaps a table) to detail the differences between Bridges and BridgesPlus, including the savings match ($) contributed - was this provided by the research or was in part of a government protection scheme? Details on retention of the sample over the 48 month and any difference by gender would have been useful to the findings. Also, was their a dose response - did the youth need to attend all the activities in both intervention approaches to receive the incentives? I know the original study has been reported elsewhere (24 months) but context is important for understanding the findings and the impact. For example, it is not clear how hopelessness is defined for the study. Further, the analysis section and results should be reviewed by a health economist or someone with stronger skills than mine for interpreting the results. I found the result as presented difficult to understand - mainly because it is not clear what the BridgesPlus received in comparison to Bridges -therefore, for me it is difficult to determine if the additional cost of BridgesPlus worth the noted impact in knowledge and hopelessness. Qualitative interviews from the participants would be very useful for a nuanced understanding of the impact on youth - would also be interesting to have a better understanding of the income-generation or educational impacts given the savings could only be used in those two area - why are these results not presented? I do think it is important that the investigators positioned the research and noting the lack of cost data for comparison. Additionally, it would have been interesting to include in discussion the use of cash-transfer or savings as government programs for protection of vulnerable youth, etc.

6. PLOS authors have the option to publish the peer review history of their article (what does this mean?). If published, this will include your full peer review and any attached files.

Reviewer #1: No

Reviewer #2: Yes: Greg Fegan

Reviewer #3: No

---

## [Author Response · Author response to Decision Letter 0]

14 Oct 2019

Reviewer #1: The manuscript is generally well written and presented. However there some minor limitations for publishing.

1. The core question being addressed is not specifically stated or clearly linked to the extensive analyses provided. This could be due to the fact that the authors assume the audience is clear in regards to the intent of the trail, as well as of he actual project itself. The authors would greatly improve this issue by providing the readers with both a slightly elaborated description of the overall project, more on the underlying rationale for a focus on cost-effectiveness of such projects in global low income and rural settings, especially those with a history of social and cultural upheaval.

We thank the reviewer for this comment. On Pages 5, Lines 24-31, we added the following explanation on the rationale of the trial and the uniqueness of FEE interventions: “Grounded in asset theory, family economic empowerment (FEE) interventions go beyond cash transfers by providing capacity building, mentorship, and seed funding and fostering household financial stability through promoting income generating activities and financial literacy. Asset theory posits that orphaned adolescents will experience higher levels of depression, have worse education outcomes, and engage in higher risk behaviors if they lack financial means to participate in secondary education. FEE interventions that bring financial stability to households act as a protective factor.”

In terms of contextualizing the importance of cost-effectiveness analysis, on Page 5, Lines 37-41, we clarified its rationale as follows: “To make the case for public investment in these interventions, it is necessary to integrate the evidence on effects with costs. By calculating an incremental cost per unit of benefit, cost-effectiveness analyses make it explicit which interventions will contribute the most relative to their costs and inform resources allocation decisions in the face of competing health priorities and resource constraints.”

Additionally, we provided more detail on the trial population, the eligibility criteria, the study setting and the study measures on Page 7 Lines 78-83; Page 7 Lines 87-88, Page 7 Lines 93-95 and Pages 8, Lines 114-127

2. The retrospective focus of the analyses does not take into account the contextual and process variables that have also influenced the observed outcomes. This is one limitation of a single paradigm (quantitative) focus and self-reported behavioral outcomes. The authors need to better address the potential influence of these factors - what was provided was highly inadequate.

Thanks for your comment. We agree with the reviewer that qualitative data would be helpful to understand contextual and process variables that have also influenced the observed outcomes, as well as the limitation in self-reported behavioral outcomes. Please refer to limitation section Page 22, Lines 369-375 where we highlighted this as “Use of self-reported measures is the most common approach to assessment in low resource settings, but this approach is limited by self-report bias. This study also relied on self-reported measures for physical and mental health and sexual risk-taking behavior. Therefore, our findings are limited to self-reported changes in these outcomes rather than a clinical diagnosis, for instance, for mental health outcomes. Future studies should consider incorporating biomarkers and other objective measures.” Also, on Page 22, Lines 365-369 we further added that “Another limitation is that because our study was quantitative by design, we did not have qualitative data on adolescent perceptions of the program, which could have added nuance and context to better elucidate the mechanisms of the effectiveness of the program on youth. Future qualitative research is needed to understand the causal pathways of effectiveness of such family economic strengthening programs.” In addition, we also want to clarify this paper presents a cost-effectiveness analysis where costs are not retrospective, but reflect the actual expenditure captured in real time.

3. In such project the personnel and setting are important to any analyses since both are critical to the functional environments (e.g, relationship with youth, physical settings, so forth). While recognized as not a part of the present study, some mention of contextual variables was both expected and needed.

Thank you for this comment. We have added details to clarify the study setting, including the eligibility criteria, that read” Those schools are in four political districts of Rakai, Masaka, Lwengo and Kalungu in South Western Uganda—a region heavily affected by HIV/AIDS. There were three eligibility criteria: 1) the adolescent had lost one or both parents to HIV/AIDS, 2) the adolescent was enrolled in grades 5 or 6, in a government-aided primary school during recruitment period, 3) the adolescent was living with a family, not an institution.” Please refer to Page 7, Lines 78-83. We also clarified the personnel for various activities on Page 7, Line 88 and Lines 93-94. 

4. Lastly, operational definitions of the key variables of health and others were not provided. This needs to be addressed.

Thank you for this comment. We have added a measurement paragraph that reads “Self-rated health is measured by a single 5-item scale ranging from excellent to very poor with higher values indicate better health. Mental health functioning is conceptualized as depression, hopelessness, self-concept and self-efficacy. We used 27-item Child Depression Inventory (CDI) and the 20-item Beck Hopelessness Scale (BHS) as operationalization with higher scores indicate worse mental health. Moreover, 20-item Tennessee Self-Concept Scale, and 29-item Youth Self-Efficacy Survey are used for measurements of self-concept and self-efficacy with higher values on indicate more positive self-concept and self-efficacy. Three indictors— sexual risk-taking intentions, HIV prevention intention, and HIV knowledge are used to capture sexual health. Detailed example questions for each scale are provided in table S2 with reliability statistics. Briefly, CDI (α = .68) and BHS (α = .65) show modest internal consistency, whereas Tennessee Self-Concept Scale, Youth Self-Efficacy, sexual risk-taking intentions, HIV prevention intention, and HIV knowledge all have acceptable internal consistency with Cronbach’s alpha greater than .70. All measurements mentioned above are standardized to be comparable across all outcomes.” Please refer to Pages 8, Lines 114-127. We also added references to S2 Table available in supplementary material. 

Overall, a good study. If the limitations are addressed this manuscript will provide a means to contribute our knowledge of such projects as suggested.

Reviewer #2: I have made comments and raised queries in the attached track-changed word documents.

However I would like to add two specific things here that I'd expect the authors to address 1) They need to note that as shown in the supplementary tables specifically Table S1 there was a marked higher attrition by 48 months for the BridgesPlus group (11.2%) as opposed to both Bridges and Control (8.8%) groups compared to the numbers at baseline. 

Thank you for this comment. We did a chi-square test and the attrition rate did not differ by study arms (χ2(2) = 2.04; p = 0.36). We have added language to the manuscript that reads “The attrition rate was 8.8% for Bridges and control group and 11.2% for the BridgesPlus group at the 48-month follow-up. However, the results from a chi-square test suggests that the attrition rates do not differ by study arms (χ2(2) = 2.04; p = 0.36). Please refer to Pages 7-8, Lines 99-112. Further, we did analyses to compare if the retained and attritted subsamples differ by demographic factors. Please refer to S2 Table, Descriptive statistics on characteristics of attritted sample in supplementary material. 

2) There should be some mention of the fact that they are using the same exposure and co-variate data to test multiple hypotheses and some readers may wonder whether there is a multiplicity problem here requiring a lower P-value to be used. Both of these should be highlighted in a small limitations sub-section within the discussion.

Thank you for this comment. We added text to the limitations paragraph addressing this specific issue that reads “A limitation of this study is that multiple hypotheses were tested for multiple outcomes using the same exposure and covariate data, potentially leading to inflated Type 1 error rates. Future studies seeking to replicate our findings might consider concentrating on the subset outcomes found to be statistically significant in this study and employing multiplicity adjustments to further control Type 1 error rates.” Please refer to Page 22, Lines 361-369. 

Reviewer #3: The investigators have presented the long-term (48 months) health-related and cost outcomes for an economic intervention with HIV/AIDs related orphans. The findings have the potential to contribute significantly to the evidence on savings-led programs with adolescents and sustained impact and cost. I have several suggestions for improvement. 

● The methods section is lacking in important details related to setting, sample, self-report measures (psychometrics, reliability of measures) used with adolescents, including data collection procedures,

Thank you for this comment. We have added details regarding the setting and the sample that read “Those schools are in four political districts of Rakai, Masaka, Lwengo and Kalungu in southwestern Uganda—a region heavily affected by HIV/AIDS. There were three eligibility criteria: 1) the adolescent had lost one or both parents to HIV/AIDS, 2) the adolescent was enrolled in grades 5 or 6, in a government-aided primary school during recruitment period, 3) the adolescent was living with a family, not an institution.” (Page 7, Lines 80-83). We also added a new sub-section on Measures that reads: “Self-rated health is measured by a single 5-item scale ranging from excellent to very poor with higher values indicate better health. Mental health functioning is conceptualized as depression, hopelessness, self-concept and self-efficacy. We used the 27-item Child Depression Inventory (CDI) and the 20-item Beck Hopelessness Scale (BHS) operationalized with higher scores indicating worse mental health. Moreover, the 20-item Tennessee Self-Concept Scale, and 29-item Youth Self-Efficacy Survey are used for measurements of self-concept and self-efficacy with higher values on indicate more positive self-concept and self-efficacy. Three indictors— sexual risk-taking intentions, HIV prevention intention, and HIV knowledge are used to capture sexual health. Detailed example questions for each scale are provided in table S2 with reliability statistics. Briefly, CDI (α = 0.68) and BHS (α = 0.65) show modest internal consistency, whereas Tennessee Self-Concept Scale, Youth Self-Efficacy, sexual risk-taking intentions, HIV prevention intention, and HIV knowledge all have acceptable internal consistency with Cronbach’s alpha greater than 0.7. All measurements mentioned above are standardized to be comparable across all outcomes.” (Pages 8, Lines 114-127). Also, please refer to S2 Table (Appendix 1, Page 5) for sample questions, psychometrics and reliability of all the measurements and Page 7, Lines 95-97 for more detail on the data collection procedures.

● and brief description (perhaps a table) to detail the differences between Bridges and BridgesPlus, including the savings match ($) contributed - was this provided by the research or was in part of a government protection scheme? 

Thank you for this comment. Table 2, on Page 16, presents the breakdown of intervention costs in all three study arms, including the matched contribution amounts in the two intervention arms. Table 2 also specifies the activities carried out in each arm and the total costs per participant. For example, it is possible to see that participants in the three arms received counselling, but only participants in the two treatment arms took part in income generating activities. 

On Page 7, Lines 98-99, we added the following clarification: “The matched funds were contributed by the trial (See Table 2 for program costs, including matched amounts per intervention arm).”

● Details on retention of the sample over the 48 month and any difference by gender would have been useful to the findings. 

Thanks for your comment. We further did analyses to compare if the retained and attritted samples differ by demographic variables. Please refer to S2 Table in Appendix 1. Descriptive statistics on characteristics of the attritted sample are given in the supplementary material. 

● Also, was their a dose response - did the youth need to attend all the activities in both intervention approaches to receive the incentives? 

Thank you for this question. The participants do not need to attend all the activities to receive the incentives. To receive the incentives, they need to 1) attend the Financial Literacy Training 2) open an account at a participating financial institution. We added these details on Page 7, Lines 95-97.

● I know the original study has been reported elsewhere (24 months) but context is important for understanding the findings and the impact. For example, it is not clear how hopelessness is defined for the study. 

Thanks for this comment. We have added a new subsection on Measures on Page 8, Lines 114-127 that reads “Self-rated health is measured by a single 5-item scale ranging from excellent to very poor with higher values indicate better health. Mental health functioning is conceptualized as depression, hopelessness, self-concept and self-efficacy. We used the 27-item Child Depression Inventory (CDI) and the 20-item Beck Hopelessness Scale (BHS) operationalized with higher scores indicating worse mental health. Moreover, the 20-item Tennessee Self-Concept Scale, and the 29-item Youth Self-Efficacy Survey are used for measurements of self-concept and self-efficacy with higher values on indicate more positive self-concept and self-efficacy. Three indictors— sexual risk-taking intentions, HIV prevention intention, and HIV knowledge are used to capture sexual health. Detailed example questions for each scale are provided in table S2 with reliability statistics. Briefly, CDI (α = 0.68) and BHS (α = 0.65) show modest internal consistency, whereas the Tennessee Self-Concept Scale, the Youth Self-Efficacy Survey, sexual risk-taking intentions, HIV prevention intention, and HIV knowledge all have acceptable internal consistency with Cronbach’s alpha greater than .70. All measurements mentioned above are standardized to be comparable across all outcomes.” We also added references to S3 Table in Appendix 1 of the supplementary material. 

● Further, the analysis section and results should be reviewed by a health economist or someone with stronger skills than mine for interpreting the results. I found the result as presented difficult to understand - mainly because it is not clear what the BridgesPlus received in comparison to Bridges -therefore, for me it is difficult to determine if the additional cost of BridgesPlus worth the noted impact in knowledge and hopelessness. 

Table 2, on Page 16, details what participants received in each of the three study arms, the breakdown of the costs per participant, including the difference in matched contributions received by Bridges and BridgesPLUS participants, as well as the total cost per participant. Specifically, Table 2 details that the program contributed a total of 20,309 Ugandan Shillings to Bridges participants and 42,634 Ugandan Shillings to BridgesPlus participants. The intervention arms, Bridges and BridgesPLUS, are described in more detail on Page 7, Lines 93-97.

● Qualitative interviews from the participants would be very useful for a nuanced understanding of the impact on youth - would also be interesting to have a better understanding of the income-generation or educational impacts given the savings could only be used in those two area - why are these results not presented? 

Thank you for your comment. We agree with the reviewer that qualitative interviews from the participants would be very useful. Unfortunately, our study was quantitative by design. We acknowledge this as a limitation on Page 22, Lines 365-369 that reads: “Another limitation is that because our study was quantitative by design, we did not have qualitative data on adolescent perceptions of the program, which could have added nuance and context to better elucidate the mechanisms of the effectiveness of the program on youth. Future qualitative research is needed to understand the causal pathways of effectiveness of such family economic strengthening programs.” 

Regarding income-generating and educational impacts, we investigated in other papers: (1) Ssewamala, F. M., Wang, J. S. H., Neilands, T. B., Bermudez, L. G., Garfinkel, I., Waldfogel, J., ... & Kirkbride, G. (2018). Cost-effectiveness of a savings-led economic empowerment intervention for AIDS-affected adolescents in Uganda: Implications for scale-up in low-resource communities. Journal of Adolescent Health, 62(1), S29-S36; (2) Wang, J. S. H., Ssewamala, F. M., Neilands, T. B., Bermudez, L. G., Garfinkel, I., Waldfogel, J., ... & You, J. (2018). Effects of Financial Incentives on Saving Outcomes and Material Well‐Being: Evidence From a Randomized Controlled Trial in Uganda. Journal of Policy Analysis and Management, 37(3), 602-629. This paper is focused on health-related outcomes and a cost-effectiveness analysis. 

● I do think it is important that the investigators positioned the research and noting the lack of cost data for comparison. 

Thank you for this comment. The lack of costing and cost-effectiveness studies on FEE interventions are highlighted in the manuscript on Page 5, Lines 37-41, and Pages 21, Lines 342-346.

● Additionally, it would have been interesting to include in discussion the use of cash-transfer or savings as government programs for protection of vulnerable youth, etc.

We agree with the reviewer. We added a full paragraph on conditional cash transfers in the discussion section on Pages 20-21 (Lines 325-339) as follows: “Cash transfer programs, both conditional and unconditional, targeted at most vulnerable households have been shown to impact a multiplicity of education and health outcomes in low resource-settings. In Malawi, the government-run cash transfer program targeting ultra-poor, labor-constrained households increased school enrollment and reduced drop-out rates and improved mental health outcomes in children after one year. In Kenya, the government-led program for orphans and vulnerable children, aimed at reducing HIV risk through unconditional cash transfers, reduced the odds of sexual debut by 31%. Structural interventions to reduce HIV incidence have been found to be the most cost-effective, particularly when targeting multiple outcomes, such as education, in addition to HIV/AIDS incidence. A privately funded program in Malawi conditioned on continued school enrollment of adolescent girls reduced the prevalence of HIV, Herpes simplex virus and depression and improved school attendance in this population. At 18 months, the intervention was estimated to avert 209 disability-adjusted life years (DALYs) at a cost of $297 per DALY averted. Despite the growing literature on cash transfer programs, few studies have focused on interventions that include an economic empowerment component.”

---

## [Decision Letter · Decision Letter 1]

9 Dec 2019

Evaluation of a savings-led family-based economic empowerment intervention for AIDS-affected adolescents in Uganda: A four-year follow-up on efficacy and cost-effectiveness

PONE-D-19-20904R1

Dear Dr. Ssewamala,

We are pleased to inform you that your manuscript has been judged scientifically suitable for publication and will be formally accepted for publication once it complies with all outstanding technical requirements.

With kind regards,

Christopher M Doran, BEc (Hons) PhD

Academic Editor

PLOS ONE

Additional Editor Comments (optional):

Reviewers' comments:

Reviewer's Responses to Questions

**Comments to the Author**

1. If the authors have adequately addressed your comments raised in a previous round of review and you feel that this manuscript is now acceptable for publication, you may indicate that here to bypass the “Comments to the Author” section, enter your conflict of interest statement in the “Confidential to Editor” section, and submit your "Accept" recommendation.

Reviewer #1: All comments have been addressed

Reviewer #2: All comments have been addressed

2. Is the manuscript technically sound, and do the data support the conclusions?

Reviewer #1: Yes

Reviewer #2: (No Response)

3. Has the statistical analysis been performed appropriately and rigorously? 

Reviewer #1: Yes

Reviewer #2: (No Response)

4. Have the authors made all data underlying the findings in their manuscript fully available?

Reviewer #1: Yes

Reviewer #2: (No Response)

5. Is the manuscript presented in an intelligible fashion and written in standard English?

Reviewer #1: Yes

Reviewer #2: (No Response)

6. Review Comments to the Author

Reviewer #1: I have read both the original submission and the updated submission with track changes. In the latter the authors provided extensive and detailed responses to each comment. The responses were clear and specific - this was expected, allowing the authors to be very targeted and strengthened the manuscript.

The oversall result was a much imporved and substantively informative manuscripr that will contrbute to our understanding of the challenges of developing and implementing programs deisgned to address wicked problems faced by children and their families in LICs and similar.

Reviewer #2: (No Response)

7. PLOS authors have the option to publish the peer review history of their article (what does this mean?). If published, this will include your full peer review and any attached files.

Reviewer #1: No

Reviewer #2: Yes: Greg Fegan

---

## [Editor Report · Acceptance letter]

12 Dec 2019

PONE-D-19-20904R1 

Evaluation of a savings-led family-based economic empowerment intervention for AIDS-affected adolescents in Uganda: A four-year follow-up on efficacy and cost-effectiveness 

Dear Dr. Ssewamala:

I am pleased to inform you that your manuscript has been deemed suitable for publication in PLOS ONE. Congratulations! Your manuscript is now with our production department. 

With kind regards,

on behalf of

Professor Christopher M Doran 

Academic Editor

PLOS ONE